# Physiological Effect of Gentle Stroking in Lambs

**DOI:** 10.3390/ani14060887

**Published:** 2024-03-13

**Authors:** Kamila Janicka, Patrycja Masier, Paulina Nazar, Patrycja Staniszewska, Grzegorz Zięba, Aneta Strachecka, Iwona Rozempolska-Rucińska

**Affiliations:** 1Institute of Biological Basis of Animal Production, University of Life Sciences in Lublin, Akademicka 13, 20-950 Lublin, Poland; patrycja.masier@up.lublin.pl (P.M.); grzegorz.zieba@up.lublin.pl (G.Z.); iwona.rucinska@up.lublin.pl (I.R.-R.); 2Department of Animal Breeding and Agricultural Consulting, University of Life Sciences in Lublin, Akademicka 13, 20-950 Lublin, Poland; paulina.nazar@up.lublin.pl; 3Department of Invertebrate Ecophysiology and Experimental Biology, University of Life Sciences in Lublin, Doświadczalna 50a, 20-280 Lublin, Poland; patrycja.staniszewska@up.lublin.pl (P.S.); aneta.strachecka@up.lublin.pl (A.S.)

**Keywords:** human-animal interaction, stroking, lamb, physiology, oxidative stress

## Abstract

**Simple Summary:**

Gentle touch has been repeatedly shown to have a positive effect on livestock. The purpose of this study was to determine potential changes in physiological indicators in lambs subjected to gentle stroking. A total of 40 lambs of two breeds were assigned to two control groups (*n* = 20) and two experimental groups (n = 20). Lambs from the experimental groups received treatment (the gentle stroking procedure). Analyses consisted of determining the following parameters: heart rate (HR), saturation (S) and biochemical analysis. An increase in cortisol levels and a decrease in total protein levels indicate a disruption of homeostasis. At the same time, a significant increase in glycogen and a significant decrease in glucose and heart rate values suggest reduced stress levels. The results obtained in the experimental groups suggest that interactions with humans, even of a positive nature, can be a source of stress, but this is a transient and adaptive reaction. Interactions with humans can be a source of stress, but long-term and positive ones will lead to positive changes and easier handling of animals.

**Abstract:**

The aim of the study was to determine changes in physiological indicators in lambs subjected to gentle stroking. The study included 40 three-week-old lambs (20 females of the synthetic prolific-meat (BCP) line and 20 females of the Świniarka (SW) breed). The animals were assigned to two control groups (*n* = 20) and two experimental groups (*n* = 20). Lambs from the experimental groups received treatment. Analyses consisted of determining the following parameters: heart rate (HR), saturation (S) and biochemical analysis. In the groups of sheep subjected to gentle massage, the analyses revealed a significant increase in the levels of cortisol, CAT, GST, GPx, among others, and a significant decrease in the levels of total protein, SOD, TAC, uric acid and Na^+^. At the same time, indicators of reduced stress levels were revealed, i.e., a significant increase in glycogen levels and a significant decrease in glucose and HR values. These results suggest that the introduction of gentle touch can induce positive states in lambs, and that the stress response may be transient and adaptive. Nevertheless, it is important to note that these interactions can be a source of stress, even if the behavioral response does not necessarily indicate stress.

## 1. Introduction

Modern animal production supported by the scientific community emphasizes the need to provide farm animals not only aversive stimulus-free conditions but also high quality of life [1,2]. This can be achieved by avoiding negative events and ensuring positive emotional states [1,3]. Human–animal interactions are an inevitable element of animal production [4], and both positive and negative experiences play a significant role in animal welfare and production [1,5]. As highlighted by Chaumont et al. [6], the creation of a positive relationship with sheep, which exhibit a particularly high level of fear of unfamiliar humans, may be quite challenging. Gentle interactions between humans and animals trigger positive behavioral responses [7,8] but do not always prove effective in some cases [3]. The perception of tactile sensations is influenced by such factors as force/power [9], mode of touch [3] and body region [7]. Tactile contact is an important part of the life of social species, as it exerts an effect on behavioral and physiological indices [3]. Sokołowski et al. [8] demonstrated that gentle physical contact plays a significant role in modeling the behavioral response in lambs from both production and primitive breeds. Gentle physical contact is associated with release of oxytocin [10], which has calming [11] and analgesic [12] effects and is responsible for the establishment of bonds and regulation of social behavior [10,13]. As a form of tactile stimulation, stroking leads to a reduction in heart rate [14]. It has been reported that cows exposed to agreeable touch have significantly lower levels of cortisol in response to stress [15], piglets cope better with changes in the environment [16] and lambs have a more effective immune system [17].

Oxidative stress is associated with the response of the organism to emotional stress [18]; hence, its markers change in stressful conditions [5]. Oxidative stress indicators are used for assessment of the balance between oxidants and antioxidants [19]. Despite their importance, there are relatively sparse data on these markers, even though they can be used to assess animal stress and welfare objectively [5]. Experiments on goats [20] and pigs [5] carried out to examine the impact of various forms of touch on behavioral and physiological indices, including oxidative stress indicators, did not reveal significant differences. It should be emphasized; however, that the study on goats did not show a negative effect of touch, and the massage protocols applied did not interfere with the homeostasis in these animals [20]. Fear is one of the breed-specific traits and is characterized by a high heritability coefficient [21]. Animals of different breeds not only have different behavioral needs but also develop different responses to environmental stimuli [22,23]. Behavioral reactions are not always indicators of stress, and in some situations it is difficult to recognize whether animals are experiencing negative or positive emotions [24]. Animals may show the same reactions under severe stress or in the absence of stress [25].

Our previous study showed that gentle touch resulted in increased curiosity and reduced fear in lambs, despite the initial indication of fear-related reactions. A question arose as to whether the observed behavioral reactions were actually related to the initially strong stress declining with the habituation to touch. In many cases, the lambs exhibited behavioral responses that could be described as “relaxation”. During the massage, they were lying calmly on the laps of the zoophysiotherapists with their eyelids closed and made no attempts to escape [8]. Since animals lack verbal expression, behavioral and physiological criteria being part of body language are the only means available to determine whether they perceive a given situation as emotionally salient information [26].

We decided that it was necessary to examine physiological indices that provide information about the reactions elicited in the lamb organism and show a potential relationship between behavioral and emotional responses.

The aim of this study was to confirm the impact of gentle touch, as a form of interaction of lambs with humans, on reduction in stress indicated by changes in physiological indicators and the behavioral reactions of these animals.

## 2. Materials and Methods

### 2.1. Animals and Housing

The research was conducted at the Small Ruminant Research Station, University of Life Sciences in Lublin, Poland. The study involved two breeds of sheep: synthetic prolific-meat line (BCP) and Świniarka (SW). The sheep were kept in a combined indoor-pasture system with constant veterinary control. The lambs were kept with mothers throughout the entire experiment period in a separated treatment pen in the sheepfold. Forty unweaned female lambs (three weeks old; *n* = 20 BCP, *n* = 20 SW) were included in the gentle stroking treatment [8] and randomly allocated to one of two groups: the control groups (C) (no massage; *n* = 10 lambs of the BCP line, *n* = 10 Świniarka breed lambs) and the experimental groups (E) (massage; *n* = 10 lambs of the BCP line, *n* = 10 Świniarka breed lambs). All the lambs came from a single pregnancy from a multiparous ewe. The average BW of SW and BCP sheep were 7.3 ± 1 kg, 8.1 ± 1.5 kg, respectively. An individual number was applied (paint-mark) to each individual to avoid misidentification.

A detailed description of the gentle touch procedure in lambs can be seen in Sokołowski et al. [8].

### 2.2. Heart Rate and Saturation

During the procedure the heart rate (HR) and saturation (S) of each lamb were measured. Once the individual was prepared (the measured area was shaved and ultrasound gel applied), the pulse oximeter was used to take measurements at the 1st and 20th minutes of the procedure. The control groups had measurements taken on the 1st and 21st days of the study (a total of four measurements), while the experimental groups had measurements taken on the 1st, 8th, 15th and 21st days (a total of eight measurements). A sensor was placed on the thoracic limb using an elastic band.

### 2.3. Blood Samples

After the massage sessions and behavioral tests had been completed (Sokołowski et al., 2023 [8]), blood samples were collected (tubes with clotting activator and containing EDTA) from the jugular vein of lambs on the 21st day after the start of the experiment. The blood samples were centrifuged at 3000 rpm (603× *g*) for 15 min in a laboratory centrifuge (MPW-350R; MPW Medical Instruments, Warsaw, Poland) at 4 °C. The samples were frozen (−20 °C) prior to subsequent analyses.

### 2.4. Biochemical Analysis

The following biochemical determinations (from the serum) were performed:
-Total protein concentrations were assayed with the Lowry et al. [27] method modified by Schaterle and Pollack [28].-The cortisol levels in the samples were determined with the General Cor (Cortisol) ELISA Kit (ELK Biotechnology CO., LTD., Denver, CO, USA). The procedures followed the manufacturer’s instructions. All samples were measured in triplets. Cortisol concentrations were expressed in ng/mL.

The antioxidant activities were measured with the following kits:Superoxide dismutase (SOD) was determined using a commercial Sigma-Aldrich (19160) SOD Determination Kit (Poznań, Poland);Catalase (CAT) was determined using a Catalase Assay Kit (219265-1KIT) from Sigma-Aldrich (Poznań, Poland);Total antioxidant capacity (TAC) was determined using a Total Antioxidant Capacity Assay Kit (MAK187-1KT) from Sigma-Aldrich (Poznań, Poland);Glutathione S-transferase (GST) activities were determined using a Sigma-Aldrich Glutathione-S-Transferase Assay Kit (CS0410);Glutathione peroxidase (GPx) activities were determined using a Glutathione Peroxidase Assay Kit from Abcam (ab102530).

All antioxidant enzyme activities were calculated per 1 mg of protein:-The concentrations of non-enzymatic antioxidants, i.e., albumin, uric acid, urea, and creatinine, were determined with the BioMaxima (Lublin, Poland) monotest.-The activities of enzymatic biomarkers such as aspartate aminotransferase (AST), alanine aminotransferase (ALT), and alkaline phosphatase (ALP) were measured with the kinetic method using monotests from Cormay (Lublin, Poland) according to the manufacturer’s procedure. Lactate dehydrogenase activities were determined using an LDH Assay Kit/Lactate Dehydrogenase Assay Kit (Colorimetric) from Abcam (Gdańsk, Poland).

The concentrations of non-enzymatic biomarkers:Triglycerides and glucose were determined with the colorimetric method using monotests from Cormay (Lublin, Poland).Sodium (Na^+^), calcium (Ca^2+^), and magnesium (Mg^2+^) ions were determined using BioMaxima (Lublin, Poland) tests according to the manufacturer’s instructions.Glycogen concentrations were measured using a Glycogen Assay Kit (K646-100, BioVision, Milpitas, CA, USA).

ATPase activities were determined with an ATPase Assay Kit from Abcam.

### 2.5. Statistical Analysis

The significance of the differences between the means was verified with the use of the multivariate analysis of variance with the least squares method in accordance with the following model:$$y_{ijk}=B_{i}+G_{j}+{\left(B\times G\right)}_{ij}+e_{ijk}$$
where: *B_i_*—the breed (BCP, Świniarka), *G_j_*—the groups (control, experimental), (*B* × *G)_ij_*—the interaction between breed and group effects and *e_ijk_*—residual error. The analyses were performed in the GLIMMIX procedure of SAS 9.4 [29].

Not all results of the breed × group interaction are shown, as the comparison of all levels with each other has no biological significance and does not influence the solution of the research problem addressed in the study.

## 3. Results

### 3.1. Differences between Breeds

Significant differences in most of the analyzed parameters were found between the breeds. Significantly higher concentrations of total protein and cortisol were determined in the SW lambs than in the BCP line. There were also significant differences in the functioning of the antioxidant system between the breeds. In comparison with the BCP breed, the Świniarka lambs exhibited significantly higher activity of antioxidant enzymes, i.e., SOD, GST and GPx, and significantly higher total antioxidant capacity (TAC). In the case of the non-enzymatic antioxidants, significantly higher albumin and urea concentrations were determined in the blood of the Świniarka lambs, while the BCP line had significantly higher creatinine concentrations. Significantly higher activity of enzymatic (AST, ALT, LDH) and non-enzymatic (triglycerides, glycogen) biomarkers and levels of Mg^2+^ ions were found in the SW breed. Significantly higher Na^+^ and Ca^2+^ concentrations were determined in the blood of the BCP lambs. The ATPase activity was significantly higher in the SW breed (Table 1). Moreover, there were significant changes in saturation (S). Its values recorded during the first measurement in the first minute of the massage (S1_1) in the SW lambs were significantly higher than the corresponding values noted in the BCP line (Table 2).

### 3.2. Differences between Groups

Significant differences were found in almost all the tested biochemical and physiological parameters in the experimental groups (E), compared to the control groups (C). The experimental groups exhibited significantly lower concentrations of total protein in the blood and a significantly higher level of cortisol. Moreover, there were significant changes in the activity of antioxidant enzymes in these groups, i.e., significantly lower SOD activity and significantly higher CAT, GST and GPx activities. The total antioxidant capacity (TAC) was significantly reduced in the massaged lambs. There were also differences in the levels of non-enzymatic antioxidants: the concentrations of albumin and urea were significantly higher, and the levels of uric acid were significantly lower in the experimental groups. The enzymatic biomarkers AST, ALP, ALT and LDH were significantly increased in the experimental groups, compared to the control animals. Additionally, differences were detected in the concentrations of non-enzymatic biomarkers and ions. The levels of triglycerides, glycogen, Ca^2+^ and Mg^2+^ were significantly higher and the concentrations of glucose and Na^+^ were significantly lower in the experimental groups. The ATPase activity was significantly reduced in the experimental animals (Table 3). Moreover, significant changes in heart rate (HR) were recorded in lambs subjected to the massage procedure (B). The values of the last measurements at the 1st and 20th minute of the massage (HR4_1, HR4_20) were significantly lower than those recorded on the first measurement day (HR1_1, HR1_20) (Table 4).

### 3.3. Differences between Groups within Breeds

The level of total protein in the sheep of the synthetic prolific-meat BCP line and the Świniarka (SW) breed was significantly lower, while the level of cortisol was significantly higher in the experimental groups in both breeds (BCP_B, SW_B), compared to the controls (BCP_K, SW_K). In the case of antioxidant enzymes, the experimental groups of both breeds had significantly lower SOD activity and significantly higher CAT, GST and GPx activities accompanied by a significantly reduced total antioxidant capacity (TAC), in comparison to the control groups. Regarding the non-enzymatic antioxidants, the experimental BCP and SW groups exhibited significantly higher albumin concentrations and significantly lower uric acid levels. The urea level was significantly higher in the massaged Świniarka lambs. The activities of enzymatic biomarkers (AST, ALP, ALT and LDH) and the concentrations of non-enzymatic biomarkers (triglycerides, glycogen) as well as Ca^2+^ and Mg^2+^ ions were increased significantly, while the levels of glucose and Na^+^ were significantly lower in the experimental groups of both breeds, compared to the control animals. The ATPase activity was significantly reduced in the experimental groups of both breeds (Table 5). Significant changes in the heart rate (HR) were recorded as well. In the BCP line, the values of the last measurement at the 1st and 20th minute of the massage (BCP_HR4_1, BCP_HR4_20) were significantly lower in the experimental groups versus the control. In the SW breed, the heart rate was significantly lower in the massaged animals at the 20th minute on the last day (SW_HR4_20), compared to the control groups (Table 6).

## 4. Discussion

Human–animal interactions elicit many behavioral and physiological changes [7], which may indicate that humans are a source of stress for animals [30]. The results obtained in the present study confirm this assumption. The first symptoms of homeostasis disturbances are changes in serum protein levels, their fractional distribution and the albumin-to-globulin ratio [31]. Therefore, it seems that the observed significant decrease in the protein level with a simultaneous increase in the albumin fraction in all the massaged lambs may evidence the presence of stress. In stress conditions, the endocrine system is stimulated as well and, consequently, the level of corticosteroids increases, which was confirmed by the increased cortisol level in the massaged animals. As indicated by Arfuso et al. [32], the plasma cortisol concentration increases in response to short-term stress associated with the sheep shearing procedure, which consists of temporary separation, capture and immobilization of the animal. The procedure applied to the experimental groups in the present study contained similar elements; hence, the animals may have been exposed to stress. However, Rault et al. [7] reported changes in cortisol concentrations in response to positive interactions, which may reflect the excitement sensation in the animal. This type of reaction cannot be ruled out in the lambs analyzed in the present study. The assessment of the behavioral indicators also showed that the massaged animals exhibited greater curiosity reflected in a greater frequency of interactions with humans [8]. The antioxidant enzyme system aims to catalyze reactions leading to the removal of free radicals [33], while oxidative stress is a result of an imbalance between the activity of antioxidants and reactive oxygen species (ROS) [34]. It has been evidenced that stressful environmental conditions may lead to excessive production of free radicals, which results in oxidative stress [30] associated with the response of the organism to emotional stress [18]. Sheep that were stroked gently were characterized by significantly lower SOD activity and total antioxidant capacity (TAC), which may indicate the development of oxidative stress [35]. However, it is worth emphasizing that the activities of CAT, GST and GPx were significantly higher in the massaged lambs. Interestingly, the activity of almost all antioxidant enzymes (SOD, GST, GPx) and the TAC value were significantly higher in the case of the Świniarka lambs. Correct interpretation of these parameters requires elucidation of their interrelationships. Superoxide dismutase (SOD) is responsible for the catalysis of the decomposition of superoxide anion (O2^−^) into hydrogen peroxide (H_2_O_2_) and oxygen (O_2_). Next, H_2_O_2_ is reduced to water (H_2_O) by catalase (CAT) and glutathione peroxidase (GPx) or another peroxiredoxin [36]. An increase in the catalase level is observed in the case of a high concentration of lipid peroxides released during SOD-mediated transformations [37] and in response to a high level of H_2_O_2_ in the organism. As reported by Szeligowska et al. [38], in response to parturition stress in ewes, the activity of glutathione S-transferase (GST) and cortisol levels increased significantly. In turn, Deger et al. [39] found a significant increase in the level of GPx with a significant decrease in the level of SOD and CAT in sheep fighting infection. It is worth nothing that the mobility of antioxidant enzymes increases in the initial period of exposure to an oxidative stress factor; hence, an increase in their activity is observed [33]. After the depletion of antioxidant enzyme cofactors (glutathione and trace elements), the activity of these enzymes declines [40]. This may explain the increased CAT, GST and GPx activities induced by the increased mobilization of these enzymes in the massaged lambs and the consequent decline in the SOD level after depletion of the cofactors. The significant changes in the concentrations of non-enzymatic antioxidants in the experimental lambs also confirm that the animals experienced oxidative stress. As reported by Venditti and Meo [35], a decrease in the concentration of non-enzymatic antioxidants indicates the presence of stress. In our study, the blood of the massaged lambs was characterized by significantly lower levels of uric acid, which is consistent, as there is a correlation between a reduced uric acid concentration and TAC [41]. In another study, a significant increase in both uric acid concentrations and TAC was found immediately after sheep shearing [42]. It should also be noted that the massaged lambs had significantly increased levels of albumin and urea. Increased ROS production may initially lead to an increase in the blood albumin concentration [41]. In turn, the increase in urea levels is probably related to a stress-induced reduction in blood flow to the kidneys. A study conducted by Autukaité et al. [43] showed a significant increase in, e.g., urea, creatinine and cortisol levels in sheep exposed to thermal stress as well as a positive correlation of cortisol with urea and protein. In the present study, the blood of all the massaged lambs exhibited a significant increase in the activity of enzymatic biomarkers, i.e., AST, ALP, ALT and LDH. It seems that the primitive breed experienced greater stress during the massage, as evidenced by the significantly higher levels of cortisol, urea and triglycerides and the significantly higher activity of enzymatic biomarkers (AST, ALT, LDH). Ineffective ROS removal may increase liver enzyme activity [33]. A significant increase in the activity of all four enzymes (AST, ALP, ALT, LDH) in response to heat stress was reported in one study [44], and a significant increase in the AST, ALT and LDH activity with a significant decrease in ALP was shown by other authors in babesiosis-infected sheep. The authors of [45,46] reported a significant increase in the activity of liver enzymes, significantly elevated concentrations of triglycerides and urea, and significantly reduced glucose levels in animals experiencing oxidative stress, which confirms the results obtained in the present study. Significantly increased triglyceride levels were found in sheep during pregnancy and after lambing [47] and in animals suffering from bluetongue disease, which additionally had increased ALT and AST activity [48]. In the present study, the blood of the massaged lambs exhibited significantly lower ATPase activity and significant changes in ion levels. Interestingly, the Świniarka lambs had higher ATPase activity and, hence, significantly lower levels of calcium ions. Calcium ATPase is responsible for removal of excess Ca^2+^ ions from the cytosol, and its activity is inhibited in stress conditions [33]. Oxidative stress is accompanied by an influx of Ca^2+^ ions from the extracellular environment into the cytoplasm [49], as observed in the massaged sheep. Casamassima et al. [50] reported significantly increased calcium and sodium levels in sheep in response to reduced water availability, which may be a source of stress for animals.

The significantly increased level of glycogen with a simultaneous decrease in glucose in the experimental groups of lambs in the present study seems to be an interesting result. It is known that, in response to stress, the process of glycogenolysis begins [51] during which glycogen is broken down to, e.g., glucose [52]. Adrenaline is the key hormone mobilizing this transformation [53]. Chronic stress has also been shown to induce a significant decrease in glycogen levels and a significant increase in glucose concentrations [54]. In contrast, different results were obtained in the groups of experimental lambs in the present study, i.e., a significant increase in the glycogen level with a simultaneous significant decrease in the glucose level, which may indicate the activation of the glycogenesis process that takes place during rest [52]. The Świniarka lambs had significantly higher concentrations of glycogen and magnesium, which may imply their greater ability to cope with stress factors. It is believed that there is a correlation between low magnesium levels and oxidative stress [55,56]. However, an increase in the Mg^2+^ concentration was found in the lambs subjected to massage in the present study. Furthermore, a high magnesium concentration prevents oxidative stress [57]. Another argument that may confirm the reduction in stress and even relaxation experienced by the animals during the massage is the reduced heart rate. The positive effect of gentle touch on lowering the heart rate in various animal species has been reported by many authors [1,3,14,58]. The present study showed that, despite the behavioral responses indicating relaxation, the lambs exposed to the gentle touch did experience stress during the human contact. The fluctuations in the physiological indicators may indicate emotional stress. However, it is worth emphasizing that this reaction may have been transient and adaptive. It has been evidenced that farm animals may perceive human presence as a threat [4]. Habituation and positive interactions with humans can cause farm animals to strive for contact that will be rewarding.

Furthermore, Serrapica et al. [59] suggests that the development of a positive relationship between lambs and stock-people based on stroking may improve the animal’s welfare. Heart rate measurements provide dynamic information about the actual animal’s state. Since a significant reduction in heart rate was demonstrated in the massaged lambs on the last day of the study, it can be expected that this interaction was perceived as a positive event. The changes in the examined indicators observed in the Świniarka breed prove a more pronounced response to the stressor, which indicates a typical response of primitive breeds that may increase the chance of survival. The behavioral response during the massage and tests also shows that the primitive breed is more fearful.

## 5. Conclusions

The present study proves that gentle touch and human contact can induce positive affective states in lambs. However, the most important conclusion of this experiment is the confirmation that the physiological reaction of the organism does not have to coincide with the behavioral reaction. Interactions with humans can be a source of stress although this was not suggested by the behavior of the lambs at the later stage of the study. Therefore, the assessment of the impact of interactions with humans on animal welfare should be based on both behavioral and physiological indicators. The present results confirm that the response to stressful factors and the rate of adaptation to these factors are breed specific, which may be a crucial element in designing the environment for animals.

Nevertheless, the results suggest further research hypotheses and problems to be solved. A question arises as to how this early tactile stimulation of lambs and the associated stress influence the ability of adult animals to cope with stressful environmental factors. There are also other questions to be answered: are the animals subjected to gentle stimulation more resistant to stress, will their behavior as adults differ from that of the control animals, and does the gentle stimulation have short- or long-term effects?

## Figures and Tables

**Table 1 animals-14-00887-t001:** Differences in the biochemical indicators between the BCP and SW breeds.

Trait	BCP		SW		*p*
	LSMEAN	SE	LSMEAN	SE	
Protein [mg/mL]	42.87	0.28	44.31	0.31	0.002
Cortisol [ng/mL]	94.30	0.33	97.26	0.36	<0.000
SOD [U/mg]	66.86	0.51	72.42	0.55	<0.000
CAT [U/mg]	15.04	0.36	13.96	0.39	0.055
GST [U/mg]	157.72	0.51	165.39	0.56	<0.000
GPx [U/mg]	44.13	0.53	48.61	0.58	<0.000
TAC [mM Trolox]	1.92	0.02	2.09	0.03	<0.000
Albumins [g/dL]	3.37	0.02	3.48	0.03	0.004
Uric acid [mg/dL]	0.34	0.00	0.32	0.00	0.057
Urea [mg/dL]	43.69	0.29	47.64	0.32	<0.000
Creatinine [mg/dL]	0.96	0.01	0.90	0.01	0.001
AST [U/L]	206.89	0.38	210.01	0.41	<0.000
ALP [U/L]	209.18	0.73	211.30	0.80	0.061
ALT [U/L]	36.20	0.35	37.56	0.38	0.013
LDH [mU/mL]	3.45	0.02	3.76	0.02	<0.000
Triglycerides [mM/L]	0.36	0.00	0.38	0.00	0.022
Glucose [mM/L]	1.93	0.03	1.97	0.03	0.355
Glycogen [µg/µL]	0.15	0.00	0.16	0.00	0.002
Na^+^ [mM/L]	134.67	0.22	132.27	0.24	<0.000
Ca^2+^ [mM/L]	2.26	0.02	1.93	0.02	<0.000
Mg^2+^ [mM/L]	1.30	0.01	1.34	0.01	<0.000
ATPase [µM/mg]	2.02	0.02	2.26	0.02	<0.000

Abbreviations: BCP—synthetic prolific-meat line breed; SW—Świniarka breed; LSMEAN—least squares mean; SE—standard error; *p*—*p*-value; SOD—superoxide dismutase; CAT—catalase; GST—glutathione S-transferase; GPx—glutathione peroxidase; TAC—total antioxidant capacity; AST—aspartate aminotransferase; ALT—alanine aminotransferase; ALP—alkaline phosphatase; LDH—lactate dehydrogenase; ATPase—adenosinetriphosphatase.

**Table 2 animals-14-00887-t002:** Differences in the heart rate and saturation between the BCP and Świniarka (SW) breeds.

Trait	Estimate BCP vs. SW	SE	Probt	AdjLower	AdjUpper
HR1_1 [bpm]	−4.75	8.69	0.589	−22.57	13.08
HR1_20 [bpm]	1.27	9.68	0.896	−18.59	21.14
HR4_1 [bpm]	7.57	6.13	0.227	−5.00	20.15
HR4_20 [bpm]	0.81	6.54	0.902	−12.61	14.23
S1_1 [%]	−5.75	2.25	0.017	−10.36	−1.13
S1_20 [%]	−4.46	3.07	0.157	−10.75	1.83
S4_1 [%]	−0.90	1.11	0.428	−3.18	1.39
S4_20 [%]	−1.34	1.54	0.390	−4.50	1.81

Abbreviations: BCP—synthetic prolific-meat line breed; SW—Świniarka breed; SE—standard error; Probt—probability test; AdjLower—adjusted lower; AdjUpper—adjusted upper; HR—heart rate; S—saturation; 1_1—measurement at the 1st minute of the massage on the first day; 1_20—measurement at the 20th minute of the massage on the first day; 4_1—measurement at the 1st minute of the massage on the last day; 4_20—measurement at the 20th minute of the massage on the last day; bpm—beats per minute.

**Table 3 animals-14-00887-t003:** Differences in the biochemical indicators between experimental (E) and control (C) groups.

Trait	BCP		SW		*p*
	LSMEAN	SE	LSMEAN	SE	
Protein [mg/mL]	41.01	0.26	46.16	0.33	<0.000
Cortisol [ng/mL]	100.20	0.30	91.36	0.39	<0.000
SOD [U/mg]	61.80	0.46	77.48	0.59	<0.000
CAT [U/mg]	17.21	0.33	11.79	0.42	<0.000
GST [U/mg]	173.81	0.47	149.30	0.60	<0.000
GPx [U/mg]	49.54	0.48	43.21	0.62	<0.000
TAC [mM Trolox]	1.76	0.02	2.25	0.03	<0.000
Albumins [g/dL]	3.57	0.02	3.29	0.03	<0.000
Uric acid [mg/dL]	0.32	0.00	0.34	0.00	<0.000
Urea [mg/dL]	46.63	0.27	44.70	0.34	<0.000
Creatinine [mg/dL]	0.92	0.01	0.94	0.01	0.262
AST [U/L]	232.81	0.35	184.09	0.44	<0.000
ALP [U/L]	217.55	0.67	202.93	0.85	<0.000
ALT [U/L]	42.00	0.32	31.77	0.40	<0.000
LDH [mU/mL]	3.67	0.02	3.53	0.02	<0.000
Triglycerides [mM/L]	0.43	0.00	0.31	0.00	<0.000
Glucose [mM/L]	1.69	0.02	2.21	0.03	<0.000
Glycogen [µg/µL]	0.18	0.00	0.13	0.00	<0.000
Na^+^ [mM/L]	131.66	0.20	135.28	0.25	<0.000
Ca^2+^ [mM/L]	2.20	0.02	1.99	0.02	<0.000
Mg^2+^ [mM/L]	1.35	0.01	1.28	0.01	<0.000
ATPase [µM/mg]	1.92	0.02	2.37	0.03	<0.000

Abbreviations: E—experimental groups; C—control groups; LSMEAN—least squares mean; SE—standard error; *p*—*p*-value; SOD—superoxide dismutase; CAT—catalase; GST—glutathione S-transferase; GPx—glutathione peroxidase; TAC—total antioxidant capacity; AST—aspartate aminotransferase; ALT—alanine aminotransferase; ALP—alkaline phosphatase; LDH—lactate dehydrogenase; ATPase—adenosinetriphosphatase.

**Table 4 animals-14-00887-t004:** Differences in the heart rate and saturation between the experimental (E) and control (C) groups.

Trait	Estimate E vs. C	SE	Probt	AdjLower	AdjUpper
HR1_1 [bpm]	−6.32	8.69	0.473	−24.14	11.51
HR1_20 [bpm]	−4.94	9.68	0.614	−24.80	14.92
HR4_1 [bpm]	−42.80	6.13	0.000	−55.38	−30.22
HR4_20 [bpm]	−53.48	6.54	0.000	−66.90	−40.05
S1_1 [%]	−0.96	2.25	0.672	−5.58	3.65
S1_20 [%]	1.90	3.07	0.540	−4.39	8.19
S4_1 [%]	−0.53	1.11	0.641	−2.81	1.76
S4_20 [%]	2.10	1.54	0.184	−1.06	5.26

Abbreviations: E—experimental groups; C—control groups; SE—standard error; Probt—probability test; AdjLower—adjusted lower; AdjUpper—adjusted upper; HR—heart rate; S—O_2_ saturation; 1_1—measurement at the 1st minute of the massage on the first day; 1_20—measurement at the 20th minute of the massage on the first day; 4_1—measurement at the 1st minute of the massage on the last day; 4_20—measurement at the 20th minute of the massage on the last day; bpm—beats per minute.

**Table 5 animals-14-00887-t005:** Differences in the biochemical indicators between the experimental (E) and control (C) groups in the BCP and Świniarka breeds.

Trait	Group E		Group C		*p*
	LSMEAN	SE	LSMEAN	SE	
Protein_BCP [mg/mL]	40.42	0.38	45.32	0.43	<0.000
Cortisol_BCP [ng/mL]	98.42	0.44	90.18	0.50	<0.000
SOD_BCP [U/mg]	58.06	0.67	75.65	0.76	<0.000
CAT_BCP [U/mg]	18.21	0.48	11.87	0.54	<0.000
GST_BCP [U/mg]	172.30	0.68	143.14	0.77	<0.000
GPx_BCP [U/mg]	46.74	0.70	41.53	0.80	<0.000
TAC_BCP [mM Trolox]	1.68	0.03	2.15	0.04	<0.000
Albumins_BCP [g/dL]	3.51	0.03	3.23	0.04	<0.000
Uric acid_BCP [mg/dL]	0.32	0.00	0.35	0.01	0.002
Urea_BCP [mg/dL]	44.18	0.39	43.20	0.44	0.107
Creatinine_BCP [mg/dL]	0.95	0.01	0.96	0.02	0.855
AST_BCP [U/L]	231.28	0.50	182.51	0.57	<0.000
ALP_BCP [U/L]	217.65	0.97	200.71	1.10	<0.000
ALT_BCP [U/L]	40.92	0.46	31.47	0.52	<0.000
LDH_BCP [mU/mL]	3.52	0.02	3.38	0.03	<0.000
Triglycerides_BCP [mM/L]	0.42	0.01	0.31	0.01	<0.000
Glucose_BCP [mM/L]	1.60	0.03	2.26	0.04	<0.000
Glycogen_BCP [µg/µL]	0.17	0.00	0.12	0.00	<0.000
Na^+^_BCP [mM/L]	132.03	0.29	137.32	0.33	<0.000
Ca^2+^_BCP [mM/L]	2.35	0.02	2.16	0.03	<0.000
Mg^2+^_BCP [mM/L]	1.34	0.01	1.26	0.01	<0.000
ATPase_BCP [µM/mg]	1.79	0.03	2.26	0.03	<0.000
Protein_SW [mg/mL]	41.60	0.36	47.01	0.51	<0.000
Cortisol_SW [ng/mL]	101.97	0.42	92.55	0.59	<0.000
SOD_SW [U/mg]	65.54	0.64	79.30	0.90	<0.000
CAT_SW [U/mg]	16.21	0.46	11.71	0.64	<0.000
GST_SW [U/mg]	175.32	0.65	155.46	0.91	<0.000
GPx_SW [U/mg]	52.34	0.67	44.88	0.94	<0.000
TAC_SW [mM Trolox]	1.84	0.03	2.34	0.04	<0.000
Albumins_SW [g/dL]	3.62	0.03	3.35	0.04	<0.000
Uric acid_SW [mg/dL]	0.31	0.00	0.34	0.01	0.006
Urea_SW [mg/dl]	49.08	0.37	46.19	0.52	0.000
Creatinine_SW [mg/dL]	0.88	0.01	0.91	0.02	0.177
AST_SW [U/L]	234.35	0.48	185.68	0.67	<0.000
ALP_SW [U/L]	217.44	0.92	205.15	1.30	<0.000
ALT_SW [U/L]	43.07	0.44	32.06	0.62	<0.000
LDH_SW [mU/mL]	3.83	0.02	3.69	0.03	0.001
Triglycerides_SW [mM/L]	0.44	0.00	0.32	0.01	<0.000
Glucose_SW [mM/L]	1.78	0.03	2.16	0.05	<0.000
Glycogen_SW [µg/µL]	0.18	0.00	0.14	0.00	<0.000
Na^+^_SW [mM/L]	131.30	0.27	133.23	0.39	<0.000
Ca^2+^_SW [mM/L]	2.04	0.02	1.83	0.03	<0.000
Mg^2+^_SW [mM/L]	1.37	0.01	1.31	0.01	<0.000
ATPase_SW [µM/mg]	2.04	0.03	2.47	0.04	<0.000

Abbreviations: E—experimental groups; C—control groups; BCP—synthetic prolific-meat line breed; SW—Świniarka breed; LSMEAN—least squares mean; SE—standard error; *p*—*p*-value; SOD—superoxide dismutase; CAT—catalase; GST—glutathione S-transferase; GPx—glutathione peroxidase; TAC—total antioxidant capacity; AST—aspartate aminotransferase; ALT—alanine aminotransferase; ALP—alkaline phosphatase; LDH—lactate dehydrogenase; ATPase—adenosinetriphosphatase.

**Table 6 animals-14-00887-t006:** Differences in the heart rate and saturation between the experimental (E) and control (C) groups in the BCP and Świniarka breeds.

Trait	Estimate E vs. C	SE	Probt	AdjLower	AdjUpper
BCP_HR1_1 [bpm]	−13.24	11.76	0.270	−45.43	18.96
BCP_HR1_20 [bpm]	−9.38	13.11	0.480	−45.26	26.50
BCP_HR4_1 [bpm]	−69.79	8.30	0.000	−92.51	−47.08
BCP_HR4_20 [bpm]	−52.95	8.86	0.000	−77.20	−28.71
BCP_S1_1 [bpm]	−2.83	3.05	0.362	−11.16	5.51
BCP_S1_20 [bpm]	0.30	4.15	0.943	−11.06	11.66
BCP_S4_1 [bpm]	−0.65	1.51	0.669	−4.78	3.48
BCP_S4_20 [bpm]	2.70	2.08	0.206	−3.01	8.40
SW_HR1_1 [%]	0.60	12.79	0.963	−34.39	35.59
SW_HR1_20 [%]	−0.50	14.25	0.972	−39.49	38.49
SW_HR4_1 [%]	−15.80	9.02	0.091	−40.49	8.89
SW_HR4_20 [%]	−54.00	9.63	0.000	−80.35	−27.65
SW_S1_1 [%]	0.90	3.31	0.788	−8.16	9.96
SW_S1_20 [%]	3.50	4.51	0.445	−8.85	15.85
SW_S4_1 [%]	−0.40	1.64	0.809	−4.88	4.08
SW_S4_20 [%]	1.50	2.27	0.514	−4.70	7.70

Abbreviations: E—experimental groups; C—control groups; BCP—synthetic prolific-meat line breed; SW—Świniarka breed; SE—standard error; Probt—probability test; AdjLower—adjusted lower; AdjUpper—adjusted upper; HR—heart rate; S—O_2_ saturation; 1_1—measurement at the 1st minute of the massage on the first day; 1_20—measurement at the 20th minute of the massage on the first day; 4_1—measurement at the 1st minute of the massage on the last day; 4_20—measurement at the 20th minute of the massage on the last day; bpm—beats per minute.

## Data Availability

The data presented in this study are available on request from the corresponding author.

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
