# Peer review of "Physiological Effect of Gentle Stroking in Lambs"

_animals, 2024, doi:10.3390/ani14060887_

Round 1

Reviewer 1 Report

Comments and Suggestions for Authors

The manuscript submitted for review entitled "Physiological effect of gentle stroking in lambs" concerns an important problem of human influence on the behavior and selected physiological and biochemical parameters of lambs' blood. 

The manuscript is written correctly, but in the Reviewer's opinion, minor changes will improve its quality

Please describe in more detail the purpose of the research conducted. It was not limited only to behavioral changes in animals.

Line 28: please correct the name of the breed

Line 51: "some cases [3 ]. The perception..." please replace "some cases [3]. The perception..."

Line 58: "regulation of social behaviour [10, 13As a form of tactile stimulation.." please replace "regulation of social behaviour [10, 13]. As a form of tactile stimulation..."

Line 123: Please provide the type and name of the device

Line 130-131: What test tubes were used to collect the blood? With a clotting activator, an anticoagulant?

Line 130: 25 or 21 day after the start of the experiment?

Line 135: Were the parameters determined in whole blood, serum or plasma?

Line 216: Please include the designation of the groups B - experimental group, K - control group in the "Material and Methods" chapter. It would be more clear to mark the experimental group as "E" and the control group as "C".

Line 235: "Abbreviations: B – research group" please replace "Abbreviations: B – experimental group"

Line 240: "Abbreviations: B – research group" please replace "Abbreviations: B – experimental group"

Line 266: check the comment for line 216

Line 268: "Abbreviations: B – research group" please replace "Abbreviations: B – experimental group"

Line 273: check the comment for line 216

Line 275: "Abbreviations: B – research group" please replace "Abbreviations: B – experimental group"

Reviewer 2 Report

Comments and Suggestions for Authors

The study by Janicka et al. investigates the physiological changes associated with positive human-animal interactions, in particular gentle stroking, in lambs of two different breeds. The improvement of the human-animal relationship is a key element of modern animal production and the authors' contribution is highly appreciated. Overall, the manuscript is well structured. The introduction, although it could be enriched in terms of background, is well structured and leads to a clear definition of the objectives. Apart from some inaccuracies, the materials and methods are well defined and the results are adequately described and discussed. The conclusions are adequately supported by the results.
Below I list some minor errors and some ideas for the implementation of the background.

Simple summary and abstract: I think it is appropriate for the authors to clearly report the number of animals allocated to each group (lines 17 and 29) and the type of treatment (line 17).
L 28 (and elsewhere in the text): I think there is a disambiguation problem, as the SW breed is sometimes referred to as Swine, and in other circumstances as Swiniarka. I ask you to please use a clear term. Thank you.
L 31: Specify the type of analysis. Thank you.
L 58: Check that the bracket for references [10,13] is closed. Thank you.
L 109: please explain what is meant by "Shantala massage", thank you.
To update the background analysis, I suggest the following references:
Line 46: https://doi.org/10.1017/S0022029920000606
Line 54 (or in the discussion): https://doi.org/10.1016/j.applanim.2016.11.007
